# UNCERTAINTY FOR DEEP IMAGE CLASSIFIERS ON OUT-OF-DISTRIBUTION DATA.

## ABSTRACT

In addition to achieving high accuracy, in many applications, it is important to estimate the probability that a model prediction is correct. Predictive uncertainty is particularly important on out-of-distribution (OOD) data where accuracy degrades. However, models are typically overconfident, and model calibration on OOD data remains a challenge. In this paper we propose a simple post hoc calibration method that significantly improves on benchmark results (Ovadia et al., 2019) on a wide range of corrupted data. Our method uses outlier exposure to properly calibrate the model probabilities.

## 1 PREDICTIVE UNCERTAINTY

When a machine learning model makes a prediction, we want to know how confident (or uncertain) we should be about the result. Uncertainty estimates are useful for both in distribution and out-of-distribution (OOD) data. Predictive uncertainty addresses this challenge by endowing model predictions with estimates of class membership probabilities. The baseline method for predictive uncertainty is to simply use the softmax probabilities of the model, $p^{\text{softmax}}(x) = \text{softmax}(f(x))$, as a surrogate for class membership probabilities (Hendrycks & Gimpel, 2017). Here $f(x)$ denotes the model outputs. Other approaches include temperature scaling (Guo et al., 2017), dropout (Gal & Ghahramani, 2016; Srivastava et al., 2014), and model ensembles (Lakshminarayanan et al., 2017), as well as Stochastic Variational Bayesian Inference (SVBI) for deep learning (Blundell et al., 2015; Graves, 2011; Louizos & Welling, 2016; 2017; Wen et al., 2018), among others.

All methods suffer from some degree of calibration error, which is the difference between predicted error rates and actual error rates, as measured by collecting data into bins based on $p_{\max} = \max_i p_i^{\text{softmax}}$ bins. The standard measurement of calibration error is the expected calibration error (ECE) Guo et al. (2017), although other measures have been used (see Nguyen et al. (2015), Hendrycks & Gimpel (2017)), including the Brier score (DeGroot & Fienberg, 1983), which is also used in Ovadia et al. (2019).

### 1.1 OUR RESULTS

We are interested in image classification problems, in particular the CIFAR-10 and Imagenet 2012 datasets, in the setting of distribution covariate shift, where the data has been corrupted by an unknown transformation of unknown intensity. These corruptions are described in Hendrycks & Dietterich (2019). Our starting point is the work of Ovadia et al. (2019) which offers a large-scale benchmark of existing state-of-the-art methods for evaluating uncertainty on classification problems under dataset shift by providing the softmax model outputs. One of main take-aways from the work of Ovadia et al. (2019) is that, unsurprisingly, the quality of the uncertainty predictions deteriorates significantly along with the dataset shift. In order to be able to calibrate for different intensity levels of *unknown* corruptions, we make use of surrogate calibration sets, which are corruptions of the data by a *different* (known) corruption. Then, when given an image (or sample of images), we first estimate the corruption level, and then recalibrate the model probabilities based on the surrogate representative calibration set. The latter step is done with a simple statistical calibration step, converting the model outputs into calibrated uncertainty estimates. Surprisingly we can estimate the corruption level just using the model outputs. We focus on the probability of correct classification, using the $p_{\max}$ values.

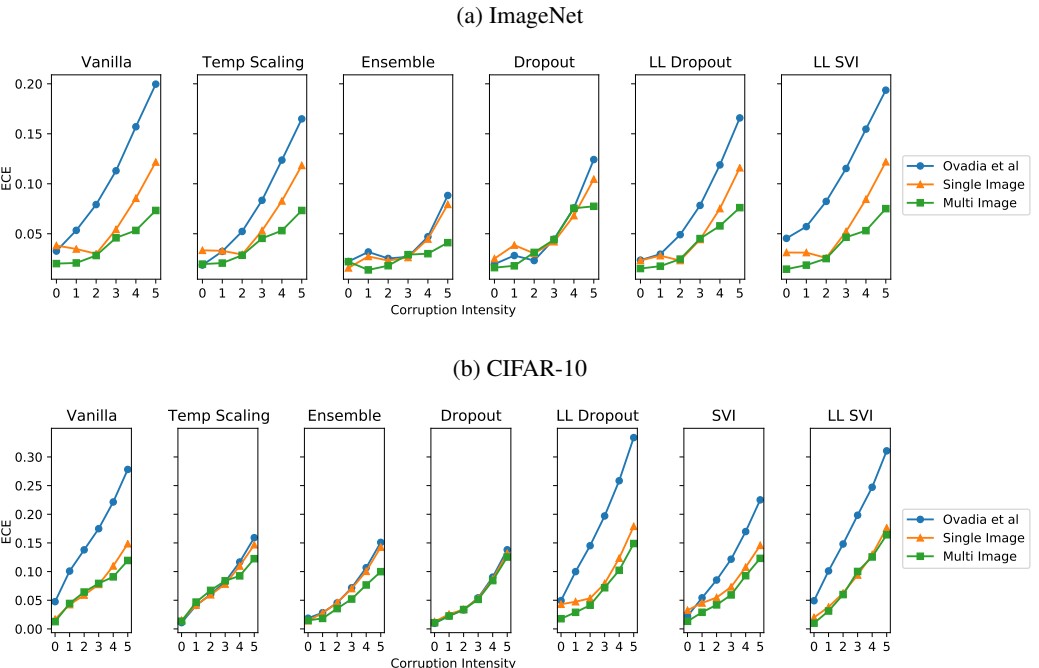

Figure 1: Comparison of the benchmark implementation (Ovadia et al., 2019), versus our single and multiple image methods. Mean Expected Calibration Error (ECE) across different corruptions types, for fixed corruption intensity going from 0 to 5. Each box represents a different uncertainty method. The ECE decreases across almost all methods and levels of intensity with the greatest improvement at the higher intensities. See Tables 1 and 2 in the Appendix for numerical comparisons.

In Figure 1 we compare each benchmark uncertainty method with our two methods. The single image method determines the appropriate calibration set using only a single image. The multiple image method uses a sample of images (all drawn from the same corruption level and type) to choose the calibration set. More images allow for a better choice of the calibration set, further reducing the calibration error. As shown in the figure, the ECE decreases across almost all methods and levels of intensity with the greatest improvement at the higher intensities. The Brier scores give similar results (see Figure 5 below). We also reproduce a figure in Ovadia et al. (2019) which gives a whisker plot of the distribution of the values.

Our results are based on the fact that data distribution shift typically leads to overconfident models (Nguyen et al., 2015): the $p_{max}$ values are *above* the true probability, and so they themselves are shifted. This allows us to use the $p_{max}$ distribution shift as a surrogate for data distribution shift and ultimately significantly reduce the calibration error using a purely *statistical approach*. In practice, we perform the model recalibration for the different corruptions and intensities based simply on the $p_{max}$ distribution shift, which we detect using surrogate corrupted calibration sets *Crucially, the corruption used to generate the surrogate corrupted datasets is left out of the test set.* The calibrated probabilities are visualized using histograms in Figure 2. In Figure 3, we can see the $p_{max}$ distributions for the chosen surrogate corrupted calibration sets and for a specific test set corruption.

## 1.2 OTHER RELATED WORK

Models trained on a given dataset are unlikely to perform as well on a shifted dataset (Hendrycks & Dietterich, 2019). Moreover, there are inevitable tradeoffs between accuracy and robustness (Chun et al., 2020). Training models against corruptions can fail to make models robust to new corruptions (Vasiljevic et al., 2016; Geirhos et al., 2018). Hendrycks et al. (2019) deals with anomaly detection, the task of distinguishing between anomalous and in-distribution data. They propose an approach called Outlier Exposure (OE) that consists in training anomaly detectors on an auxiliary dataset

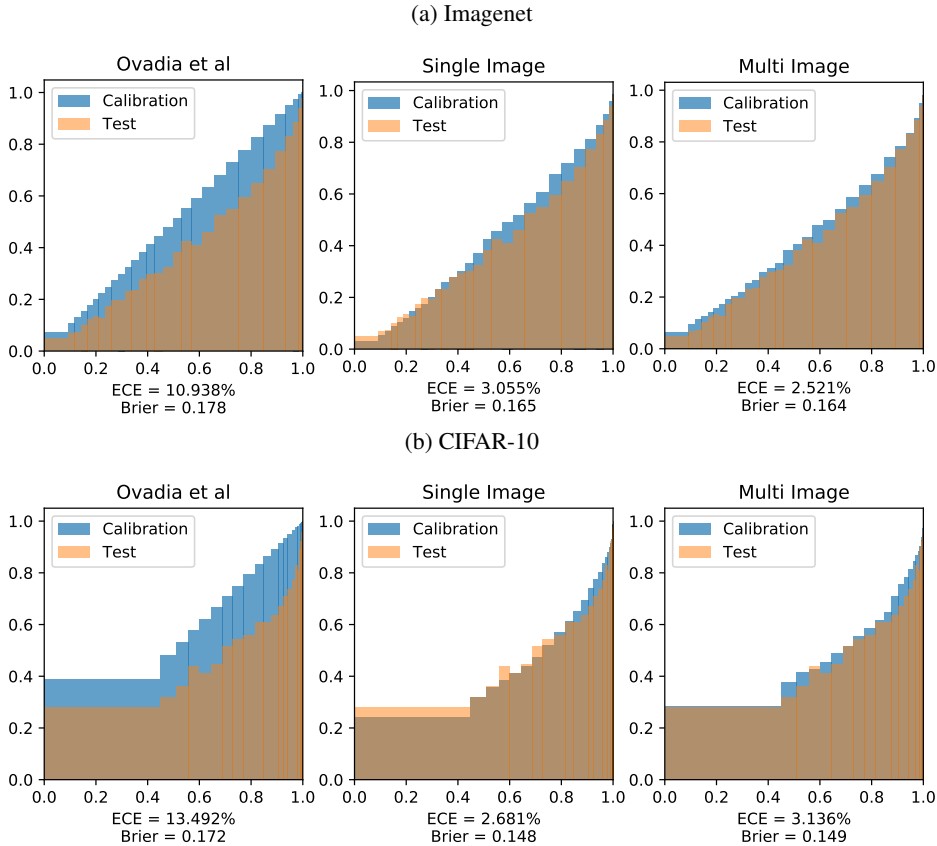

Figure 2: Visualization of calibration errors for the Vanilla method on corrupted images on ImageNet and CIFAR-10 using the elastic transform corruption with intensity 4. The x-axis corresponds to the $p_{max}$ values and the y-axis to the confidence estimates of $p^{correct}$. The blue histogram is the calibration probabilities, the orange is the test probabilities, and the brown is where both overlap. The ECE is large (lower is better) for the Vanilla method at higher corruption levels, due to the probability shift. The Brier score (lower is better) is also improved. The gap between the orange and blue curves represents the calibration error. Notice that we used 30 equally sized bins, so in the CIFAR-10 plot, there are very few values below .4, which is why the first bin is wide.

of outliers. Similarly in Hendrycks et al. (2020), the authors propose AUGMIX, a method that improves both robustness and uncertainty measures by exposing the model to perturbed images during training. Shao et al. (2020) propose a confidence calibration method that uses an auxiliary class to separate mis-classified samples from correctly classified ones which thus allowing the mis-classified samples to be assigned with a low confidence. Nado et al. (2020) argues that the internal activations of the deep models also suffer from covariate shift in the presence of OOD images. Thus they propose to recompute the batch norm at prediction time using a sample of the unlabeled images from the test distribution improving the accuracy and ultimately the calibration. Park et al. (2020) and Wang et al. (2020) focus on the more general problem of unsupervised domain adaptation where one assumes assumes to have unlabeled examples from the test distribution which may only share the same classification classes as the train distribution. Park et al. (2020) propose an approach based on importance weighting to correct for the covariate shift in the data, together with learning an indistinguishable feature map between training and test distributions. Wang et al. (2020) extend the temperature scaling method into domain adaption achieving more accurate calibrations with lower bias and variance without introducing any hyperparameters.

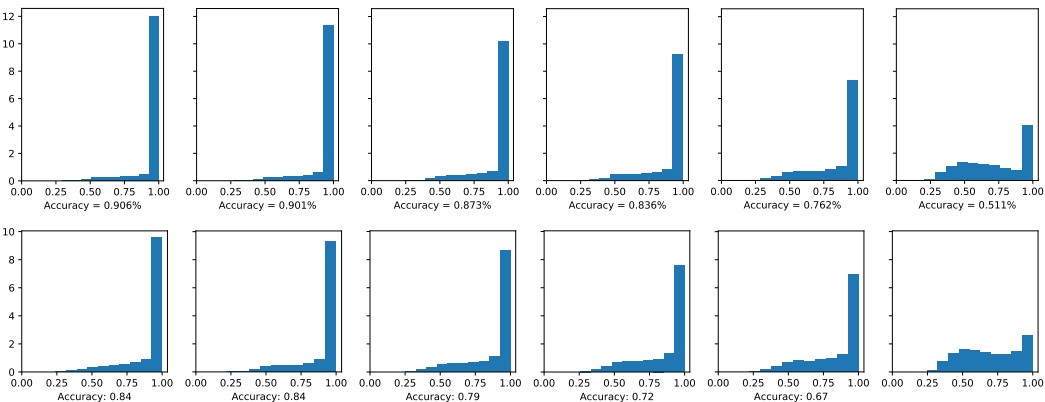

Figure 3: Histogram of the $p_{\max}$ values for the vanilla method on CIFAR-10. Top: calibration sets $\{0\}$, $\{0,1\}$, $\{0,2\}$, $\{0,3\}$, $\{0,4\}$, and $\{0,5\}$ (left to right) with contrast corruption. Bottom: test set with elastic transform corruption with increased intensities ranging from 1 to 5 (left to right) and SVHN dataset (right most picture). The multi image method outputs calibrated probabilities based the surrogate calibration set $\{0,3\},\{0,3\},\{0,3\},\{0,4\},\{0,4\},\{0,5\}$, respectively.

## 2 BACKGROUND

### 2.1 CLASSIFICATION AND LABEL PROBABILITIES

Predictive uncertainty seeks to estimate the probability that an input $x$ belongs to each class,

$$p_k^{class}(x) = \mathbb{P}\left[y = k \mid x\right], \qquad \text{for each } k \in \mathcal{Y}. \tag{1}$$

Here we write $x \in \mathcal{X}$ for data, $y \in \mathcal{Y} = \{1, \ldots, K\}$ for labels.

In the benchmark methods, the softmax of the model outputs, $p^{\text{softmax}}(x) = \text{softmax}(f(x))$ are used as a surrogate for the class probabilities. In the case of the vanilla and temperature scaling methods, $f(x)$ is simply the model outputs. Similarly, for the Ensemble or Dropout methods, $p^{\text{softmax}}(x)$ represents the average of the probability vectors over the multiple models or queries of the model, respectively.

Generally speaking, these softmax values are not an accurate prediction of the class probabilities $p_k^{class}(x)$ (Domingos & Pazzani, 1996). Here we focus on the *correct classification*

$$p^{\text{correct}}(x) = \mathbb{P}\left[y = \hat{y}(x)\right] \tag{2}$$

where the classification of the model is given by $\hat{y}(x) = \arg\max f_i(x)$. Guo et al. (2017) showed that $p_{\max} = \max_i p_i^{\text{softmax}}$ usually overestimates $p^{\text{correct}}$. We can extend our method to top 5 correctness, as well as to other quantities of interest, such as, using different binning methods, based on structured predictors (Kuleshov & Liang, 2015).

## 3 METHOD

**Problem definition** We are given a model (or ensemble of models) trained on a given dataset $\rho_{\text{train}}$. We want to have calibrated error estimates on an unknown (different) data set $\rho_{\text{test}}$, which could have different levels of corruption. Since one model cannot be calibrated on all of the possible (different) datasets, we want to allow for multiple calibrations, and apply them accordingly.

We study two cases: (i) we have a single image drawn from an unknown distribution, or (ii) we have multiple images, each drawn from the same unknown distribution. In the latter case, we use the full test set (we obtain similar results using 100 images).

### 3.1 CALIBRATING FOR DATASET SHIFT

We choose $C$ distinct calibration sets generated from shifted distributions $\rho^{\text{CAL},j}$. These sets are chosen to have different representative degrees of corruption intensity. Each calibration set leads

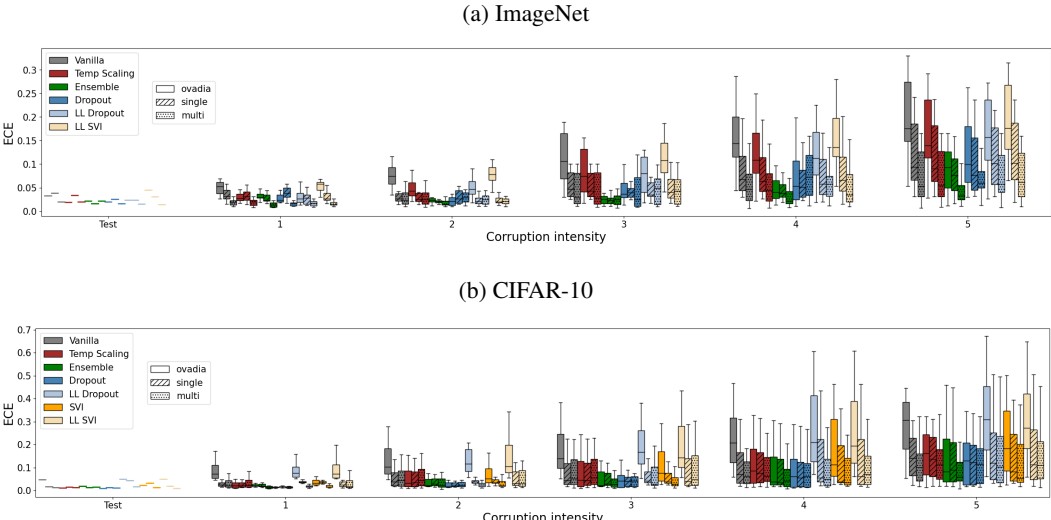

Figure 4: Comparison of the benchmark implementation (Ovadia et al., 2019) versus our single and multiple image methods. Expected Calibration Error (ECE) distribution across different corruptions types, for fixed corruption intensity going from 0 to 5. Each box represents a different uncertainty method.

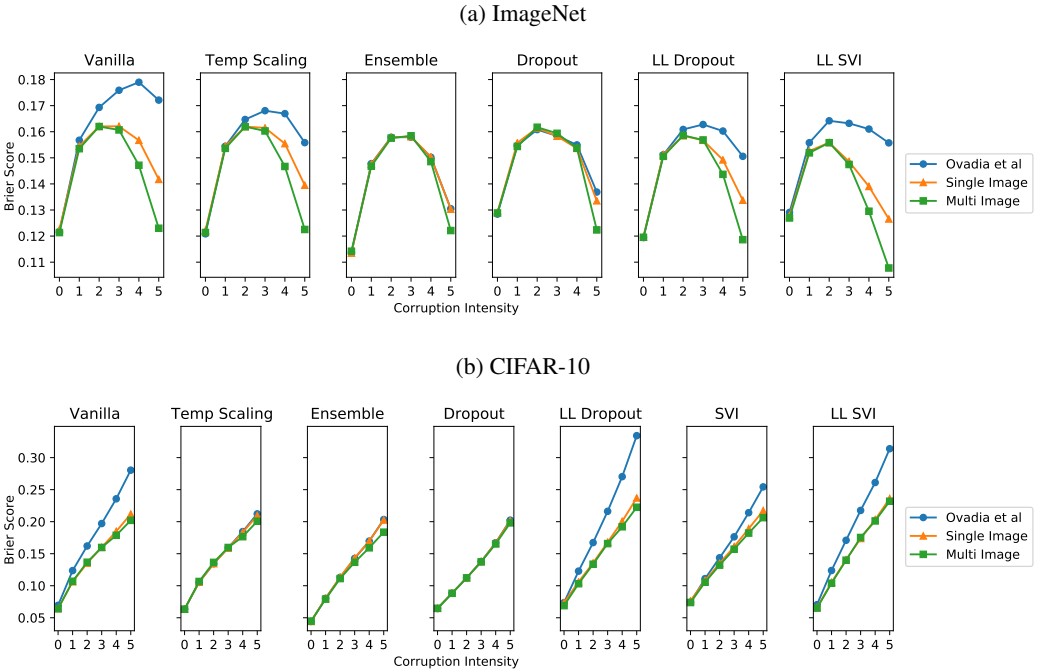

Figure 5: Comparison of the benchmark implementation (Ovadia et al., 2019) versus our single and multiple image methods. Mean Brier score across different corruptions types, for fixed corruption intensity going from 0 to 5. Each box represents a different uncertainty method. See Tables 3 and 4 for numerical comparisons.

to a different uncertainty estimate for a given $p_{\max}$ value. We adaptively choose the calibration set given an image (Single Image Method) or a test set of images (Multiple Image Method).

(a) ImageNet

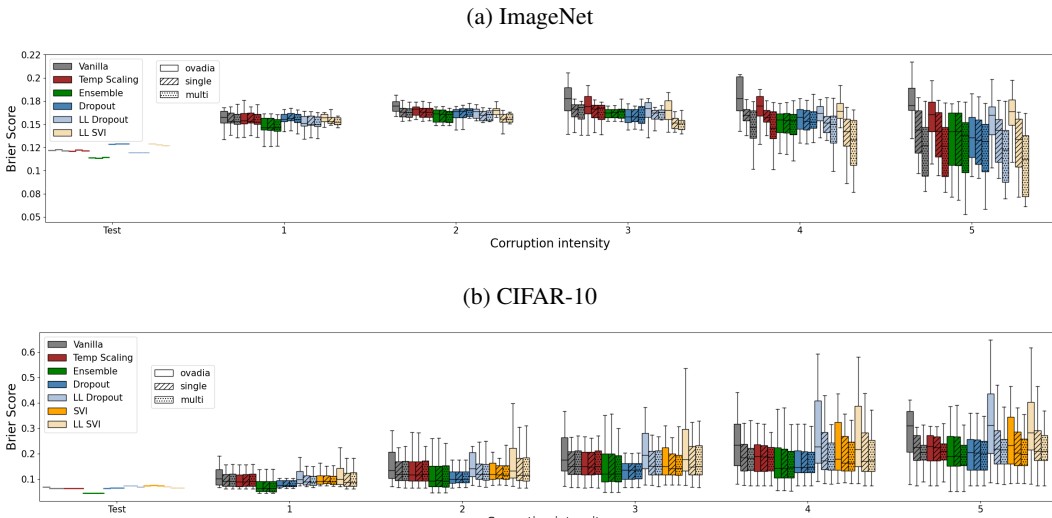

(b) CIFAR-10

Figure 6: Comparison of the benchmark implementation (Ovadia et al., 2019) versus our single and multiple image methods. Brier score distribution across different corruptions types, for fixed corruption intensity going from 0 to 5. Each box represents a different uncertainty method. See Tables 3 and 4 in the Appendix for numerical comparisons.

For each calibration dataset $S^{\mathrm{CAL},j}$, we convert the $p_{\max}$ values on that calibration set into the $p_{\mathrm{correct}}$ values, as follows. Recall that $p_{\max} = \max_i p_i^{\mathrm{softmax}}$, i.e., the model's probability for the predicted class.

(i) Evaluate the model $p_{\max}$ values on the calibration set, $P^{\mathrm{CAL},j} = \{p_{\max}(x) \mid x \in S^{\mathrm{CAL},j}\}$. Record the probability density $h^{\mathrm{CAL},j}(x)$, of the $p_{\max}$ values as a histogram, by binning the $p_{\max}$ using equally spaced bins.

(ii) Define the calibrated model probabilities by
$$p_{\mathrm{correct}}^{\mathrm{CAL},j}(x) = \mathbb{P}\left[y = \hat{y}(x) \mid p_{\max}(x)\right], \tag{3}$$
using ground truth labels on the calibration set $S^{\mathrm{CAL},j}$. These probabilities are computed using a histogram in the following way. Partition $S^{\mathrm{CAL},j} = \{x_1, \ldots, x_m\}$ into bins $B_i$ according its $p_{\max}$ values. Given an image $x$ with $p_{\max}(x) \in B_i$, approximate $p_{\mathrm{correct}}^{\mathrm{CAL},j}(x)$ as
$$\frac{1}{|B_i|} \sum_{p_{\max}(x_j) \in B_i} \mathbf{1}_{y(x_j) = \hat{y}(x_j)}$$
See (Oberman et al., 2020) for more details.

**Single Image Method:** Given a single image $x$ drawn from an unknown distribution $\rho_{\mathrm{test}}$, we estimate the likelihood that the corruption level of the image corresponds to each of the calibration sets, and then take the corresponding weighted average of the calibrated probabilities. Ideally, in the single image method we would like to obtain $q_i(p_{\max})$ close to one for the calibration set whose $p_{\max}$ distribution looks closely to the $p_{\max}$ distribution of the test images.

(i) The probability that the $p_{\max}$ value came from a given calibration set, making the standard assumption that the *a priori* likelihoods of the calibration sets are all equal, is
$$q_i(p_{\max}) = \frac{h^{\mathrm{CAL},i}(p_{\max})}{\sum_{j=1}^{C} h^{\mathrm{CAL},j}(p_{\max})}$$

(ii) Then the calibrated probability, conditional on each of the calibration sets, is given by
$$p_{\mathrm{correct}}^{\mathrm{test}}(x) = \sum_{j=1}^{C} q_j(p_{\max}(x)) \, p_{\mathrm{correct}}^{\mathrm{CAL},j}(x)$$

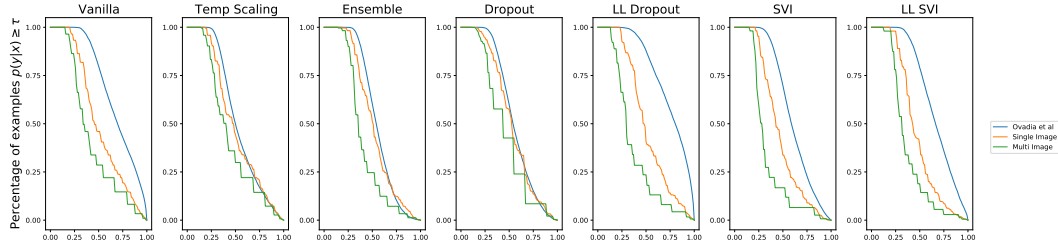

Figure 7: Confidence of CIFAR-10 (bottom) trained models on entirely OOD data (SVHN dataset (Netzer et al., 2011)). The benchmark method (blue) has highest confidence. The single and multi image method are much less confidence on OOD data.

**Multiple Image Method:** Given a sample of images $S^{\text{test}} = \{x_1, \ldots, x_m\}$ drawn from an unknown distribution, $\rho_{\text{test}}$, where $m > 1$.

(i) Record the corresponding model $p_{\max}$ values, $P^{\text{test}} = \{p_{\max}(x) \mid x \in S^{\text{test}}\}$, and compute the mean $\mu$.

(ii) Compare to the means $\mu_j$ of $P^{\text{CAL}}$ and find the closest mean for the calibration sets, $i = \arg\min_j |\mu - \mu_j|$. Set

$$p_{\text{correct}}^{\text{test}}(x) = p_{\text{correct}}^{\text{CAL},i}(x).$$

We can use a simpler formula for the multiple image method, because with multiple samples, knowing the mean is sufficient to estimate the correct calibration set.

### 3.2 Practical Implementation of Our Method

In Ovadia et al. (2019) the additional methods used include (LL) Approximate Bayesian inference for the parameters of the last layer only (Riquelme et al., 2018), (LL SVI) Mean field stochastic variational inference on the last layer, (LL Dropout) Dropout only on the activations before the last layer. We refer to Ovadia et al. (2019) for more details on how each method was implemented. All these methods ultimately use the softmax probabilities as a surrogate for the class probabilities. The difference between the methods is how these probabilities are obtained.

The distributional shift on ImageNet used 16 corruption types with corruption intensity on a scale from 0 to 5: various forms of Noise, and Blur as well as Pixelate, Saturate, Brightness, Contrast, Fog and Frost, etc. See Figure S3 in (Ovadia et al., 2019).

We used the published softmax value of each of the methods from the benchmark dataset in Ovadia et al. (2019). We selected Contrast as the corruption to use for calibration, *removing it from the test set*. We calibrated our models using equally sized bins, on each of following calibration sets: $\{0\}$, $\{0,1\}$, $\{0,2\}$, $\{0,3\}$, $\{0,4\}$, and $\{0,5\}$. For instance the set $0, 1$ corresponds to clean images and respective corruption with an intensity level of 1. Heuristically we always want clean images in our calibration set while having different shifted means as a result of increasingly corrupted images (see Figure 3). Without the clean images, the single image method would become uncalibrated for in-distribution images as the calibration sets would have a disproportional amount of corrupted images. For both CIFAR-10 and Imagenet, we select 5000 images for calibration, which are shared across the different calibration sets. This means that for instance the calibration set corresponding to $\{0, 1\}$ contains a total of 10000 images: the selected 5000 clean images and their corrupted counterparts at an intensity level of 1. A more sophisticated combination of calibration sets could lead to improvements.

We reproduced the figures in Ovadia et al. (2019) with a small adjustment: we measure the ECE and Brier score just for $p_{\text{correct}}$, rather than for $p_k^{class}$. However this made a negligible difference to the values. In addition, ECE can depend on the binning procedures (equally spaced or equally sized). Equally sized bins are more effective for calibration since they reduce statistical error. They do however lead to different bin edges on different calibration sets, which required combining the bins. This can be done by refinement (which we used here) or simple by one dimensional density estimation (Wasserman, 2006, Chapter 6).

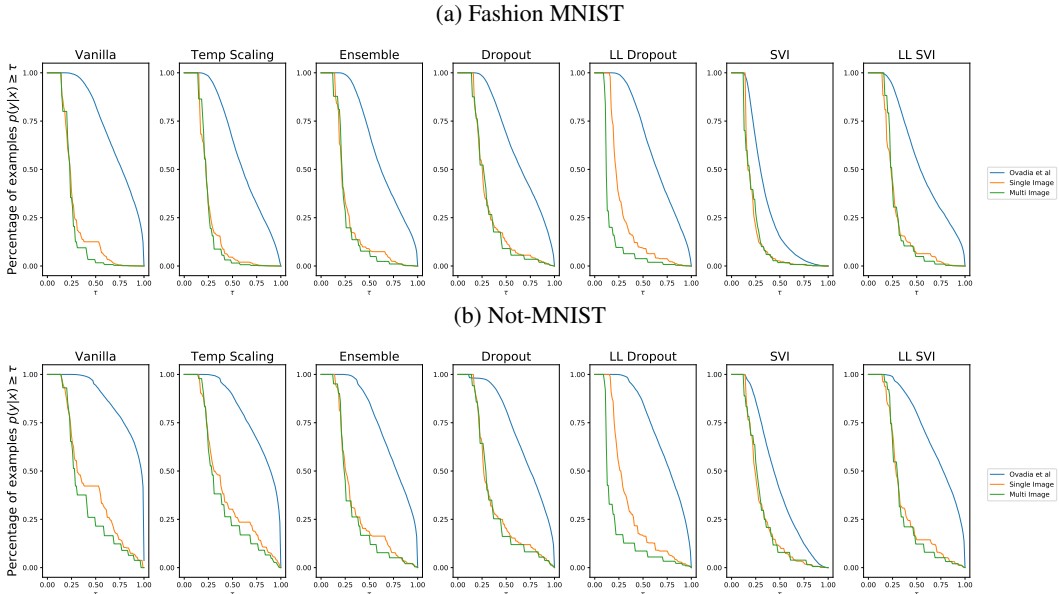

Figure 8: Confidence of MINST trained models on entirely OOD data: Fashion-MNIST (Xiao et al., 2017) and Not-MNIST (Bulatov, 2011). Our proposed methods are significantly less confident on OOD than the benchmark method.

### 3.3 DISCUSSION OF THE RESULTS

Both our single and multi image methods improve on the benchmark widely across methods and corruption intensity levels, as shown in Figure 1 and Tables 1, and 2, which report the mean ECE scores for each model and across different corruption types, for fixed corruption intensity going from 0 to 5. The multi image method performs better than the single image with a few exceptions. This is a natural consequence of using more images to better estimate the corruption level. While the ensemble method remains overall the best method, the gap to other methods is significantly reduced.

Moreover, we test the OOD detection performance. We evaluate the CIFAR-10 trained models on the SVHN dataset (Netzer et al., 2011), in addition to MNIST trained models (for these we used rotation to form the surrogate calibration sets) on Fashion-MNIST (Xiao et al., 2017) and Not-MNIST (Xiao et al., 2017). Ideally, the models should not be confident when presented with this completely OOD data. As we can see in Figures 7 and 8, both the single and multi images methods result in methods that are significantly less confident when compared to the benchmark data (Ovadia et al., 2019). We can explain the increased performance by looking at the $p_{\max}$ distributions depicted in Figure 3. For the SVHN dataset, the multi image method recalibrated the model based on calibration set $\{0, 5\}$: the dataset shift is correctly captured by the $p_{\max}$ shift and the probabilities are recalibrated accordingly. By exposing the model to corrupted images at the calibration stage, it now "knows what it does not know". We note that instead of requiring class predictions for all classes, our method only requires $p_{\max}$ values, which is a considerably smaller data set. Moreover, we compile these values into a histogram with 30 bins, making the added cost of our method negligible.

### 4 CONCLUSIONS

Increasingly we are asking models trained on a given dataset to perform on out of distribution data. Our work focused on uncertainty estimates, in particular, an estimate of the probability that our model classification is correct. In contrast to most deep uncertainty work, we use a purely statistical approach to reduce the calibration error of deep image classifiers under dataset shift. The approach is model agnostic, so it can be applied to future models.

Previous work has shown that uncertainty estimates degrade on corrupted data, as measured by the expected calibration error. The greater the mismatch between training data and test data, the greater

the degradation of uncertainty estimates. We overcome this limitation introducing a method which allows a given model to be better calibrated to multiple corruption intensities. Our method works by no longer requiring that model outputs approximate class probabilities. We add a simple extra calibration step, and detect the level of corruption of data, which allows the use calibrations tuned to the corruption level of the data.

Table 1: Comparison on Imagenet of the benchmark implementation (Ovadia et al., 2019) versus our single and multiple image methods. Numerical values of the means of ECE scores across different corruptions types, for fixed corruption intensity going from 0 to 5.

| | | | | Corruption Intensity | | | |
|---|---|---|---|---|---|---|---|
| | Method | Test | 1 | 2 | 3 | 4 | 5 |
| Vanilla | (Ovadia et al.) | 0.0327 | 0.0534 | 0.0792 | 0.1130 | 0.1571 | 0.1997 |
| | (Single Image) | 0.0382 | 0.0348 | 0.0300 | 0.0543 | 0.0855 | 0.1218 |
| | (Multi Image) | **0.0201** | **0.0208** | **0.0282** | **0.0459** | **0.0533** | **0.0733** |
| Temp Scaling | (Ovadia et al.) | **0.0187** | 0.0323 | 0.0523 | 0.0835 | 0.1237 | 0.1649 |
| | (Single Image) | 0.0334 | 0.0329 | 0.0292 | 0.0531 | 0.0828 | 0.1183 |
| | (Multi Image) | 0.0196 | **0.0207** | **0.0285** | **0.0454** | **0.0531** | **0.0733** |
| Ensemble | (Ovadia et al.) | 0.0222 | 0.0318 | 0.0253 | 0.0269 | 0.0470 | 0.0883 |
| | (Single Image) | **0.0158** | 0.0273 | 0.0232 | **0.0262** | 0.0447 | 0.0793 |
| | (Multi Image) | 0.0221 | **0.0138** | **0.0181** | 0.0291 | **0.0301** | **0.0411** |
| Dropout | (Ovadia et al.) | 0.0196 | 0.0282 | **0.0232** | 0.0446 | 0.0759 | 0.1242 |
| | (Single Image) | 0.0251 | 0.0386 | 0.0303 | **0.0420** | **0.0681** | 0.1046 |
| | (Multi Image) | **0.0161** | **0.0179** | 0.0314 | 0.0441 | 0.0755 | **0.0775** |
| LL Dropout | (Ovadia et al.) | 0.0236 | 0.0296 | 0.0491 | 0.0783 | 0.1189 | 0.1659 |
| | (Single Image) | 0.0233 | 0.0280 | **0.0232** | **0.0442** | 0.0752 | 0.1160 |
| | (Multi Image) | **0.0152** | **0.0176** | 0.0246 | 0.0452 | **0.0579** | **0.0761** |
| LL SVI | (Ovadia et al.) | 0.0454 | 0.0571 | 0.0824 | 0.1153 | 0.1547 | 0.1937 |
| | (Single Image) | 0.0312 | 0.0312 | 0.0258 | 0.0522 | 0.0845 | 0.1220 |
| | (Multi Image) | **0.0146** | **0.0186** | **0.0252** | **0.0465** | **0.0531** | **0.0751** |

Table 2: Comparison on CIFAR-10 of the benchmark implementation Ovadia et al. (2019) versus our single and multiple image methods. Numerical values of the means ECE scores across different corruptions types, for fixed corruption intensity going from 0 to 5.

| | | | | Corruption Intensity | | | |
|---|---|---|---|---|---|---|---|
| | Method | Test | 1 | 2 | 3 | 4 | 5 |
| Vanilla | (Ovadia et al.) | 0.0475 | 0.1009 | 0.1379 | 0.1748 | 0.2214 | 0.2782 |
| | (Single Image) | 0.0170 | **0.0423** | **0.0590** | **0.0776** | 0.1098 | 0.1485 |
| | (Multi Image) | **0.0127** | 0.0444 | 0.0643 | 0.0794 | **0.0908** | **0.1194** |
| Temp Scaling | (Ovadia et al.) | **0.0110** | **0.0410** | 0.0603 | 0.0822 | 0.1167 | 0.1591 |
| | (Single Image) | 0.0150 | 0.0415 | **0.0595** | **0.0781** | 0.1097 | 0.1468 |
| | (Multi Image) | 0.0134 | 0.0468 | 0.0672 | 0.0839 | **0.0925** | **0.1225** |
| Ensemble | (Ovadia et al.) | 0.0187 | 0.0283 | 0.0453 | 0.0718 | 0.1068 | 0.1509 |
| | (Single Image) | **0.0141** | 0.0277 | 0.0458 | 0.0703 | 0.1005 | 0.1427 |
| | (Multi Image) | 0.0149 | **0.0183** | **0.0355** | **0.0521** | **0.0766** | **0.0999** |
| Dropout | (Ovadia et al.) | **0.0097** | 0.0228 | **0.0331** | 0.0542 | 0.0903 | 0.1380 |
| | (Single Image) | 0.0127 | 0.0257 | 0.0339 | 0.0531 | 0.0868 | 0.1319 |
| | (Multi Image) | 0.0108 | **0.0225** | 0.0338 | **0.0515** | **0.0843** | **0.1254** |
| LL Dropout | (Ovadia et al.) | 0.0494 | 0.0999 | 0.1452 | 0.1971 | 0.2584 | 0.3338 |
| | (Single Image) | 0.0430 | 0.0474 | 0.0533 | 0.0792 | 0.1233 | 0.1789 |
| | (Multi Image) | **0.0178** | **0.0289** | **0.0414** | **0.0718** | **0.1022** | **0.1489** |
| SVI | (Ovadia et al.) | 0.0224 | 0.0542 | 0.0853 | 0.1216 | 0.1698 | 0.2251 |
| | (Single Image) | 0.0326 | 0.0452 | 0.0547 | 0.0733 | 0.1077 | 0.1458 |
| | (Multi Image) | **0.0133** | **0.0290** | **0.0419** | **0.0593** | **0.0928** | **0.1231** |
| LL SVI | (Ovadia et al.) | 0.0491 | 0.1011 | 0.1481 | 0.1984 | 0.2472 | 0.3106 |
| | (Single Image) | 0.0200 | 0.0382 | 0.0626 | **0.0940** | 0.1295 | 0.1763 |
| | (Multi Image) | **0.0100** | **0.0312** | **0.0599** | 0.1002 | **0.1255** | **0.1642** |

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

## A    ABLATION AND CROSS VALIDATION STUDY

We start by investigating the impact of the choice of corruption for the calibration set. Ideally, the choice of corruption should be representative of the distribution of corruptions, so a mild corruption or a very strong corruption would give slightly worse results. At the same time, here we demonstrate that choosing a different corruption should not significantly degrade the results.

In Figure 9 we perform a cross-validation study over the choice of corruption used to generate the calibration sets (always leaving it out of the corruptions used at test time). We plot the mean and variance of the ECE across different validation corruptions types.

For CIFAR-10, both the single and multi image method are robust to the choice of validation corruption. On ImageNet, the multi image method the improvement is consistent across the validation corruption chosen except for the dropout method. As for the single image method the calibration at lower levels of intensity is degraded using certain corruptions, for example the glass blur corruption. However, this seems to be caused by the strength of the corruption: the accuracy on glass blur level 1 was roughly half that of clean images. We hypothesize that better results could be obtained by simply having the corruption strength be proportional to the loss of accuracy, as is the case of the contrast corruption (see Figure 3) In practice, the choice of corruption for single image method should be such that the method remains calibrated for in-distribution images.

We investigate as well the impact of adding corrupted images to the calibration sets. In order to do so, we compare the results of our proposed method and the benchmark from Ovadia et al. (2019) with the calibration obtained from using a single calibration set with only clean images like the method proposed in (Oberman et al., 2020). We refer to it as the Top1 binning method. As we can see in Figure 10, if the classifier is not well-calibrated, e.g., the vanilla or the LL SVI classifiers, there is a consistent improvement across all corruption intensity levels, with the improvement being only marginal for ImageNet. Moreover, when the classifier is well-calibrated, Top1 binning does not improve calibration (e.g. the Temp Scaling classifier) or even decreases it at high levels of corruption intensity (e.g. Ensemble and Dropout).

Finally, we explore why the method works in practice. Figure 11 shows us that without any calibration the ECE scores become higher when the mismatch between the $p_{\max}$ distribution of the training set and the $p_{\max}$ distribution of the test set increases. Here we measure the mismatch in terms of the $p_{\max}$ means, the same criteria used in the multi image method. These qualitative results are confirmed by the Pearson's correlation coefficient. This correlation justifies why detecting the $p_{\max}$ distribution shit allows us to significantly improve the calibration of the different methods: in practice our proposed methods perform the recalibration of the model based on the calibration set whose $p_{\max}$ distribution is closest to the $p_{\max}$ distribution of the test set. Moreover, one notices the higher the correlation, the bigger the calibration improvement provided by both our single and multi image methods. For instance, Dropout has the lowest Pearson's $r$ score and it is also the method where we notice the least improvement. On the other hand, Vanilla has the largest improvement and also the highest Pearson's $r$ score.

## B    TABLES OF BRIER METRICS

Table 3 and Table 4 report the mean Brier scores for each model and dataset across different corruption types, for fixed corruption intensity going from 0 to 5. The Brier scores can be computed directly from the data, without binning. The ranking provided by the Brier scores is quite similar that provided by the ECE. On ImageNet the only difference is Ensemble at corruption level 1. On CIFAR-10 there were two ranking differences.

## C    TABLES OF ECE METRICS ACCROSS DIFFERENT CORRUPTIONS

Table 5 and Table 6 report the ECE scores for the vanilla model across different corruption types and intensities ranging from 0 to 5 for ImageNet and CIFAR-10, respectively. Contrast is the corruption used for to form the calibration sets.

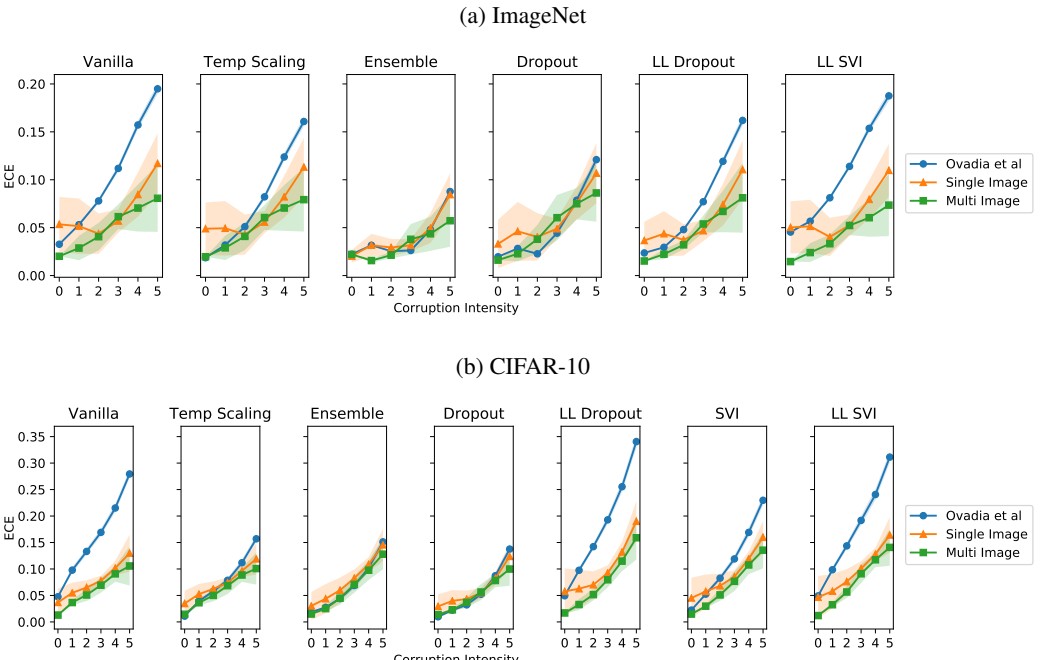

Figure 9: Figure 1 shows us the mean ECE across different corruptions types, for fixed corruption intensity going from 0 to 5 when contrast is used in the calibration sets. Here we show how those means change when different corruptions are used in the calibration set. For CIFAR-10, our proposed methods are robust to the choice of corruption used in the calibration set, while for ImageNet the choice of the corruption is import, in particular for the single image method.

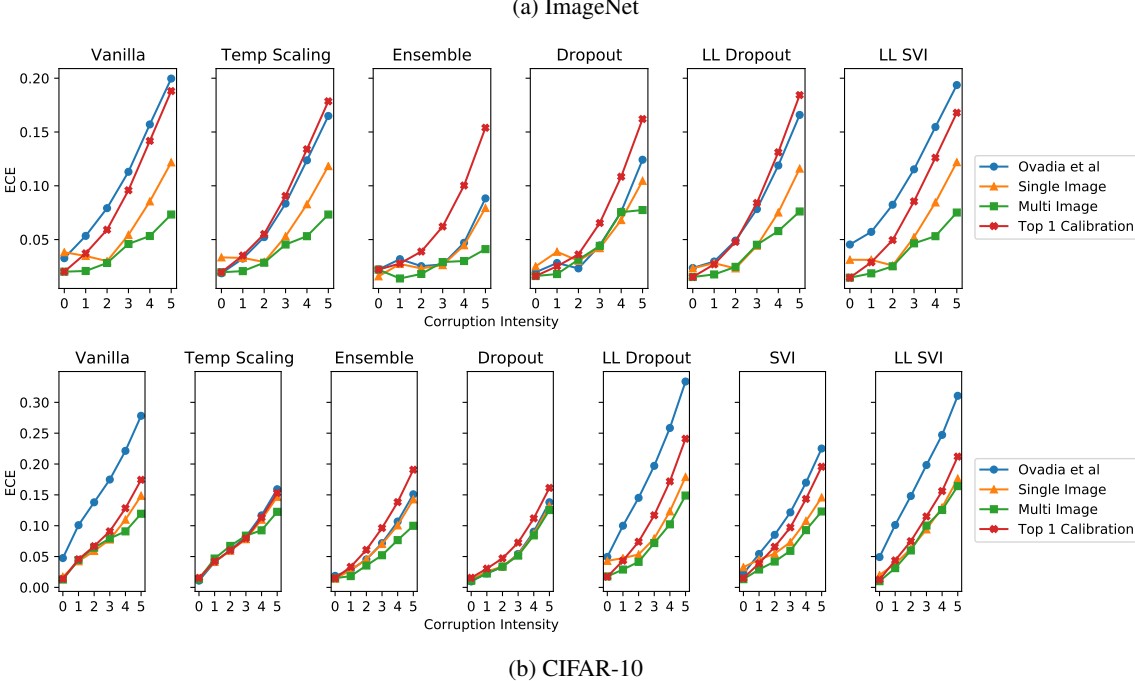

Figure 10: Comparison of the benchmark implementation (Ovadia et al., 2019), versus our proposed single and multiple image methods and Top1 binning (Oberman et al., 2020) for CIFAR-10. Mean Expected Calibration Error (ECE) across different corruptions types, for fixed corruption intensity going from 0 to 5.

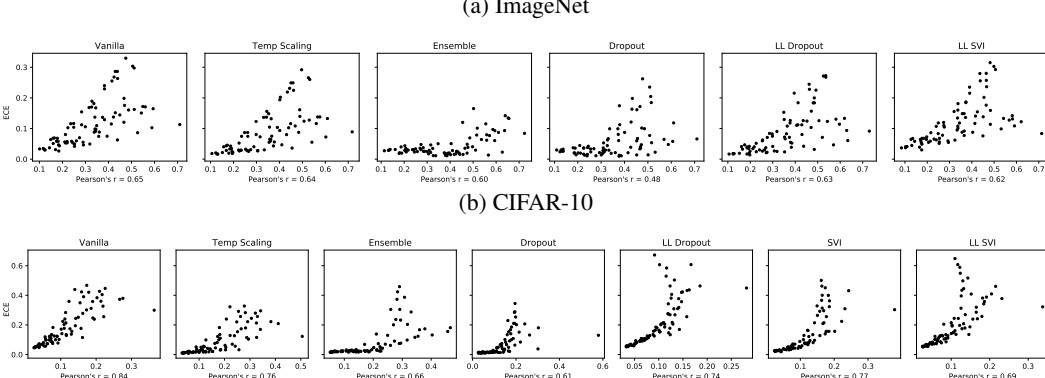

Figure 11: ECE (pre-calibration) versus $|\mu^{\text{test}} - \mu^{\text{train}}|$ and corresponding Pearson's $r$ score, where $\mu^{\text{test}}, \mu^{\text{train}}$ denote the $p_{\max}$ mean of the test and training set, respectively. Each point in the plot represents a different corruption at a different level of intensity.

Table 3: Comparison on Imagenet of the benchmark implementation (Ovadia et al., 2019) versus our single and multiple image methods. Numerical values of the means of the Brier scores across different corruptions types, for fixed corruption intensity going from 0 to 5.

| Method | | Test | Corruption Intensity | | | | |
| --- | --- | --- | --- | --- | --- | --- | --- |
| | | | 1 | 2 | 3 | 4 | 5 |
| Vanilla | (Ovadia et al.) | 0.1217 | 0.1567 | 0.1693 | 0.1759 | 0.1790 | 0.1721 |
| | (Single Image) | 0.1225 | 0.1545 | 0.1621 | 0.1621 | 0.1567 | 0.1417 |
| | (Multi Image) | **0.1213** | **0.1535** | **0.1620** | **0.1607** | **0.1472** | **0.1230** |
| Temp Scaling | (Ovadia et al.) | **0.1209** | 0.1544 | 0.1647 | 0.1681 | 0.1669 | 0.1558 |
| | (Single Image) | 0.1221 | 0.1545 | 0.1620 | 0.1615 | 0.1554 | 0.1395 |
| | (Multi Image) | 0.1214 | **0.1536** | **0.1619** | **0.1603** | **0.1467** | **0.1225** |
| Ensemble | (Ovadia et al.) | 0.1136 | 0.1478 | 0.1579 | **0.1579** | 0.1502 | 0.1305 |
| | (Single Image) | **0.1135** | 0.1476 | 0.1579 | 0.1581 | 0.1504 | 0.1303 |
| | (Multi Image) | 0.1142 | **0.1468** | **0.1575** | 0.1584 | **0.1485** | **0.1221** |
| Dropout | (Ovadia et al.) | **0.1284** | 0.1547 | **0.1608** | 0.1584 | 0.1550 | 0.1369 |
| | (Single Image) | 0.1290 | 0.1557 | 0.1615 | **0.1583** | **0.1536** | 0.1335 |
| | (Multi Image) | 0.1289 | **0.1544** | 0.1617 | 0.1594 | 0.1537 | **0.1224** |
| LL Dropout | (Ovadia et al.) | **0.1194** | 0.1512 | 0.1609 | 0.1628 | 0.1603 | 0.1506 |
| | (Single Image) | 0.1197 | 0.1513 | **0.1585** | **0.1567** | 0.1492 | 0.1337 |
| | (Multi Image) | 0.1195 | **0.1506** | 0.1586 | 0.1568 | **0.1437** | **0.1186** |
| LL SVI | (Ovadia et al.) | 0.1291 | 0.1558 | 0.1642 | 0.1632 | 0.1610 | 0.1557 |
| | (Single Image) | 0.1278 | 0.1526 | 0.1559 | 0.1486 | 0.1390 | 0.1266 |
| | (Multi Image) | **0.1269** | **0.1519** | **0.1558** | **0.1475** | **0.1295** | **0.1078** |

Table 4: Comparison of CIFAR-10 of the benchmark implementation Ovadia et al. (2019) versus our single and multiple image methods. Numerical values of means Brier scores across different corruptions types, for fixed corruption intensity going from 0 to 5.

| Method | | Test | Corruption Intensity | | | | |
| | | | 1 | 2 | 3 | 4 | 5 |
|---|---|---|---|---|---|---|---|
| Vanilla | (Ovadia et al.) | 0.0694 | 0.1236 | 0.1620 | 0.1971 | 0.2357 | 0.2804 |
| | (Single Image) | 0.0644 | **0.1061** | **0.1352** | 0.1600 | 0.1848 | 0.2120 |
| | (Multi Image) | **0.0640** | 0.1070 | 0.1367 | **0.1597** | **0.1787** | **0.2021** |
| Temp Scaling | (Ovadia et al.) | **0.0633** | **0.1053** | **0.1345** | 0.1593 | 0.1845 | 0.2122 |
| | (Single Image) | 0.0636 | 0.1054 | 0.1345 | **0.1588** | 0.1833 | 0.2096 |
| | (Multi Image) | 0.0634 | 0.1066 | 0.1364 | 0.1597 | **0.1766** | **0.2007** |
| Ensemble | (Ovadia et al.) | 0.0443 | 0.0803 | 0.1129 | 0.1426 | 0.1696 | 0.2033 |
| | (Single Image) | **0.0442** | 0.0803 | 0.1135 | 0.1433 | 0.1697 | 0.2026 |
| | (Multi Image) | 0.0447 | **0.0791** | **0.1112** | **0.1366** | **0.1592** | **0.1836** |
| Dropout | (Ovadia et al.) | **0.0644** | 0.0883 | **0.1122** | 0.1378 | 0.1672 | 0.2024 |
| | (Single Image) | 0.0647 | 0.0885 | 0.1123 | 0.1376 | 0.1668 | 0.2014 |
| | (Multi Image) | 0.0648 | **0.0882** | 0.1123 | **0.1373** | **0.1652** | **0.1981** |
| LL Dropout | (Ovadia et al.) | 0.0736 | 0.1228 | 0.1673 | 0.2162 | 0.2703 | 0.3343 |
| | (Single Image) | 0.0735 | 0.1060 | 0.1357 | 0.1679 | 0.2005 | 0.2370 |
| | (Multi Image) | **0.0689** | **0.1031** | **0.1336** | **0.1658** | **0.1920** | **0.2225** |
| SVI | (Ovadia et al.) | 0.0744 | 0.1111 | 0.1436 | 0.1763 | 0.2140 | 0.2543 |
| | (Single Image) | 0.0765 | 0.1081 | 0.1349 | 0.1610 | 0.1890 | 0.2173 |
| | (Multi Image) | **0.0738** | **0.1055** | **0.1322** | **0.1570** | **0.1822** | **0.2063** |
| LL SVI | (Ovadia et al.) | 0.0704 | 0.1238 | 0.1710 | 0.2176 | 0.2611 | 0.3139 |
| | (Single Image) | 0.0658 | 0.1053 | 0.1407 | **0.1737** | 0.2032 | 0.2361 |
| | (Multi Image) | **0.0652** | **0.1039** | **0.1398** | 0.1753 | **0.2012** | **0.2319** |

Table 5: Comparison of ImageNet of the benchmark implementation Ovadia et al. (2019) versus our single and multiple image methods for the vanilla classifier. Numerical values of ECE scores for different corruptions at different intensity levels going from 0 to 5. The contrast corruption was used to form the calibration sets.

| | | | Corruption Intensity | | | | |
|---|---|---|---|---|---|---|---|
| Corruption | | Test | 1 | 2 | 3 | 4 | 5 |
| Brightness | (Ovadia et al.) | 0.0327 | 0.0333 | 0.0348 | 0.0371 | 0.0436 | 0.0533 |
| | (Single Image) | 0.0382 | 0.0436 | 0.0425 | 0.0427 | 0.0396 | 0.0315 |
| | (Multi Image) | **0.0201** | **0.0209** | **0.0225** | **0.0106** | **0.0059** | **0.0073** |
| Defocus Blur | (Ovadia et al.) | 0.0327 | 0.0417 | 0.0494 | 0.0628 | 0.0862 | 0.1026 |
| | (Single Image) | 0.0382 | 0.0483 | 0.0392 | **0.0325** | **0.0350** | **0.0423** |
| | (Multi Image) | **0.0201** | **0.0192** | 0.0376 | 0.0950 | 0.0493 | 0.0533 |
| Elastic Transform | (Ovadia et al.) | 0.0327 | 0.0266 | 0.0881 | 0.0595 | 0.1094 | 0.2633 |
| | (Single Image) | 0.0382 | 0.0576 | 0.0183 | 0.0257 | 0.0306 | 0.1745 |
| | (Multi Image) | **0.0201** | **0.0208** | **0.0173** | **0.0109** | **0.0252** | **0.0904** |
| Fog | (Ovadia et al.) | 0.0327 | 0.0531 | 0.0699 | 0.0972 | 0.1288 | 0.1985 |
| | (Single Image) | 0.0382 | 0.0324 | 0.0211 | 0.0302 | 0.0497 | 0.1139 |
| | (Multi Image) | **0.0201** | **0.0143** | **0.0194** | **0.0149** | **0.0465** | **0.0378** |
| Frost | (Ovadia et al.) | 0.0327 | 0.0528 | 0.0985 | 0.1375 | 0.1515 | 0.1755 |
| | (Single Image) | 0.0382 | 0.0308 | 0.0252 | 0.0578 | 0.0699 | 0.0935 |
| | (Multi Image) | **0.0201** | **0.0099** | **0.0180** | **0.0298** | **0.0158** | **0.0204** |
| Gaussian Blur | (Ovadia et al.) | 0.0327 | 0.0311 | 0.0464 | 0.0750 | 0.1206 | 0.1708 |
| | (Single Image) | 0.0382 | 0.0527 | **0.0405** | 0.0333 | 0.0494 | 0.0943 |
| | (Multi Image) | **0.0201** | **0.0161** | 0.0407 | **0.0138** | **0.0331** | **0.0404** |
| Gaussian Noise | (Ovadia et al.) | 0.0327 | 0.0693 | 0.1025 | 0.1684 | 0.2552 | 0.2976 |
| | (Single Image) | 0.0382 | **0.0155** | **0.0272** | 0.0814 | 0.1659 | 0.2121 |
| | (Multi Image) | **0.0201** | 0.0262 | 0.0472 | **0.0804** | **0.0780** | **0.1436** |
| Glass Blur | (Ovadia et al.) | 0.0327 | 0.0495 | 0.0745 | 0.1624 | 0.1720 | 0.1647 |
| | (Single Image) | 0.0382 | 0.0354 | 0.0220 | 0.0837 | 0.0956 | 0.0930 |
| | (Multi Image) | **0.0201** | **0.0105** | **0.0123** | **0.0184** | **0.0372** | **0.0379** |
| Impulse Noise | (Ovadia et al.) | 0.0327 | 0.1157 | 0.1543 | 0.1892 | 0.2681 | 0.3035 |
| | (Single Image) | 0.0382 | **0.0337** | 0.0666 | **0.1002** | 0.1783 | 0.2171 |
| | (Multi Image) | **0.0201** | 0.0595 | **0.0638** | 0.1003 | **0.0910** | **0.1466** |
| Pixelate | (Ovadia et al.) | 0.0327 | 0.0640 | 0.0684 | 0.1042 | 0.1371 | 0.1367 |
| | (Single Image) | 0.0382 | 0.0209 | **0.0171** | **0.0284** | 0.0534 | **0.0522** |
| | (Multi Image) | **0.0201** | **0.0207** | 0.0236 | 0.0483 | **0.0511** | 0.0541 |
| Saturate | (Ovadia et al.) | 0.0327 | 0.0512 | 0.0583 | 0.0298 | 0.0561 | 0.1121 |
| | (Single Image) | 0.0382 | 0.0311 | 0.0264 | 0.0494 | 0.0299 | 0.0327 |
| | (Multi Image) | **0.0201** | **0.0095** | **0.0138** | **0.0177** | **0.0091** | **0.0237** |
| Shot Noise | (Ovadia et al.) | 0.0327 | 0.0682 | 0.1165 | 0.1814 | 0.2863 | 0.3297 |
| | (Single Image) | 0.0382 | **0.0190** | **0.0353** | **0.0931** | 0.1971 | 0.2420 |
| | (Multi Image) | **0.0201** | 0.0243 | 0.0609 | 0.0941 | **0.1124** | **0.1654** |
| Spatter | (Ovadia et al.) | 0.0327 | 0.0309 | 0.0574 | 0.1056 | 0.1674 | 0.2401 |
| | (Single Image) | 0.0382 | 0.0464 | 0.0266 | 0.0273 | 0.0808 | **0.1503** |
| | (Multi Image) | **0.0201** | **0.0136** | **0.0099** | **0.0196** | **0.0804** | 0.1536 |
| Speckle Noise | (Ovadia et al.) | 0.0327 | 0.0587 | 0.0804 | 0.1702 | 0.2294 | 0.2863 |
| | (Single Image) | 0.0382 | 0.0235 | **0.0142** | 0.0826 | **0.1395** | 0.1963 |
| | (Multi Image) | **0.0201** | **0.0158** | 0.0248 | **0.0819** | 0.1440 | **0.1093** |
| Zoom Blur | (Ovadia et al.) | 0.0327 | 0.0553 | 0.0885 | 0.1149 | 0.1443 | 0.1605 |
| | (Single Image) | 0.0382 | **0.0303** | 0.0274 | **0.0467** | 0.0678 | 0.0807 |
| | (Multi Image) | **0.0201** | 0.0310 | **0.0117** | 0.0531 | **0.0207** | **0.0157** |

Table 6: Comparison of CIFAR-10 of the benchmark implementation Ovadia et al. (2019) versus our single and multiple image methods for the vanilla classifier. Numerical values of ECE scores for different corruptions at different intensity levels going from 0 to 5. The contrast corruption was used to form the calibration sets.

| Corruption | | Test | Corruption Intensity | | | | |
|---|---|---|---|---|---|---|---|
| | | | 1 | 2 | 3 | 4 | 5 |
| Brightness | (Ovadia et al.) | 0.0475 | 0.0465 | 0.0482 | 0.0541 | 0.0596 | 0.0697 |
| | (Single Image) | 0.0170 | 0.0180 | 0.0191 | 0.0178 | 0.0192 | 0.0233 |
| | (Multi Image) | **0.0127** | **0.0151** | **0.0113** | **0.0147** | **0.0161** | **0.0184** |
| Defocus Blur | (Ovadia et al.) | 0.0475 | 0.0478 | 0.0526 | 0.0874 | 0.1282 | 0.2555 |
| | (Single Image) | 0.0170 | 0.0150 | 0.0177 | **0.0193** | 0.0348 | 0.0919 |
| | (Multi Image) | **0.0127** | **0.0120** | **0.0133** | 0.0214 | **0.0336** | **0.0418** |
| Elastic Transform | (Ovadia et al.) | 0.0475 | 0.0735 | 0.0732 | 0.1004 | 0.1349 | 0.1737 |
| | (Single Image) | 0.0170 | 0.0211 | 0.0199 | 0.0287 | **0.0268** | **0.0553** |
| | (Multi Image) | **0.0127** | **0.0168** | **0.0144** | **0.0282** | 0.0314 | 0.0648 |
| Fog | (Ovadia et al.) | 0.0475 | 0.0459 | 0.0473 | 0.0579 | 0.0749 | 0.1946 |
| | (Single Image) | 0.0170 | 0.0187 | 0.0239 | 0.0284 | 0.0265 | **0.0675** |
| | (Multi Image) | **0.0127** | **0.0123** | **0.0133** | **0.0142** | **0.0140** | 0.0787 |
| Frost | (Ovadia et al.) | 0.0475 | 0.0810 | 0.1336 | 0.2240 | 0.2401 | 0.3589 |
| | (Single Image) | 0.0170 | **0.0267** | **0.0499** | **0.1083** | **0.1258** | **0.2255** |
| | (Multi Image) | **0.0127** | 0.0282 | 0.0623 | 0.1211 | 0.1354 | 0.2423 |
| Gaussian Blur | (Ovadia et al.) | 0.0475 | 0.0482 | 0.0899 | 0.1565 | 0.2468 | 0.3729 |
| | (Single Image) | 0.0170 | 0.0145 | **0.0183** | **0.0436** | 0.0904 | 0.1910 |
| | (Multi Image) | **0.0127** | **0.0115** | 0.0240 | 0.0529 | **0.0296** | **0.0949** |
| Gaussian Noise | (Ovadia et al.) | 0.0475 | 0.1712 | 0.2781 | 0.3833 | 0.4266 | 0.4466 |
| | (Single Image) | 0.0170 | **0.0627** | **0.1349** | 0.2253 | 0.2632 | 0.2786 |
| | (Multi Image) | **0.0127** | 0.0744 | 0.1556 | **0.1439** | **0.1716** | **0.1872** |
| Glass Blur | (Ovadia et al.) | 0.0475 | 0.4228 | 0.3902 | 0.3492 | 0.4673 | 0.4207 |
| | (Single Image) | 0.0170 | **0.2732** | **0.2379** | **0.2034** | **0.3107** | **0.2643** |
| | (Multi Image) | **0.0127** | 0.2936 | 0.2593 | 0.2234 | 0.3323 | 0.2868 |
| Impulse Noise | (Ovadia et al.) | 0.0475 | 0.1226 | 0.2027 | 0.2518 | 0.3267 | 0.3794 |
| | (Single Image) | 0.0170 | **0.0475** | **0.0906** | **0.1197** | 0.1663 | 0.2073 |
| | (Multi Image) | **0.0127** | 0.0556 | 0.1031 | 0.1366 | **0.0980** | **0.1143** |
| Pixelate | (Ovadia et al.) | 0.0475 | 0.0653 | 0.0975 | 0.1331 | 0.2836 | 0.4390 |
| | (Single Image) | 0.0170 | **0.0179** | **0.0349** | **0.0513** | **0.1676** | **0.3039** |
| | (Multi Image) | **0.0127** | 0.0187 | 0.0419 | 0.0659 | 0.1826 | 0.3220 |
| Saturate | (Ovadia et al.) | 0.0475 | 0.0559 | 0.0713 | 0.0507 | 0.0658 | 0.0995 |
| | (Single Image) | 0.0170 | 0.0245 | **0.0216** | 0.0170 | 0.0267 | **0.0296** |
| | (Multi Image) | **0.0127** | **0.0194** | 0.0249 | **0.0117** | **0.0214** | 0.0395 |
| Shot Noise | (Ovadia et al.) | 0.0475 | 0.1082 | 0.1769 | 0.3012 | 0.3579 | 0.4046 |
| | (Single Image) | 0.0170 | **0.0360** | **0.0741** | **0.1546** | 0.2041 | 0.2406 |
| | (Multi Image) | **0.0127** | 0.0433 | 0.0811 | 0.1759 | **0.1232** | **0.1510** |
| Spatter | (Ovadia et al.) | 0.0475 | 0.0726 | 0.1052 | 0.1253 | 0.1193 | 0.1828 |
| | (Single Image) | 0.0170 | 0.0244 | **0.0395** | **0.0443** | **0.0488** | **0.0897** |
| | (Multi Image) | **0.0127** | **0.0236** | 0.0463 | 0.0514 | 0.0614 | 0.1083 |
| Speckle Noise | (Ovadia et al.) | 0.0475 | 0.1155 | 0.2138 | 0.2440 | 0.3118 | 0.3601 |
| | (Single Image) | 0.0170 | **0.0339** | **0.0985** | **0.1176** | 0.1626 | 0.1992 |
| | (Multi Image) | **0.0127** | 0.0475 | 0.1109 | 0.1337 | **0.0930** | **0.1102** |
| Zoom Blur | (Ovadia et al.) | 0.0475 | 0.0817 | 0.1022 | 0.1374 | 0.1765 | 0.2387 |
| | (Single Image) | 0.0170 | 0.0249 | 0.0225 | **0.0199** | 0.0447 | 0.0846 |
| | (Multi Image) | **0.0127** | **0.0235** | **0.0197** | 0.0297 | **0.0606** | **0.0322** |
| Translation | (Ovadia et al.) | 0.0475 | 0.0554 | 0.1240 | 0.1410 | 0.1224 | 0.0540 |
| | (Single Image) | 0.0170 | 0.0186 | **0.0410** | **0.0421** | **0.0385** | 0.0237 |
| | (Multi Image) | **0.0127** | **0.0147** | 0.0477 | 0.0453 | 0.0488 | **0.0176** |

