# OpenReview forum: "Uncertainty for deep image classifiers on out of distribution data. "
_ICLR.cc/2021/Conference — Reject_

### Official Review · AnonReviewer4 · 2020-10-16
**New method for preventing OOD detection in image classification, but unclear when and why method works**

**Rating:** 6
**Confidence:** 4

**Review:**

Thank you for the additional experiments. Especially, Figure 7 and 8 look promising.
My conclusion from the experiments is that the "contrast" corruption (used for validation) seems to be general enough, in the sense that for many other corruptions, encountered at test time, the performance is good.
However, as AnonReviewer1, I am not sure about why the methodology seems to work well for very different types corruptions at test time, and completely OOD data (Figure 7 and 8).
More empirical/theoretical analysis would be nice.
Increased rating to 6.

------

Summary:
The paper addresses the important problem of over-confident predictions on out-of distribution (OOD) data.
Focusing on image classification, they propose to calibrate probabilities on a validation dataset which contains images that were artificially corrupted in various ways (noise, blur, brightness etc.). This is different from traditional post-hoc methods which use an uncorrupted validation data set (iid assumption). This way they can explicitly calibrate probabilities for the non-iid assumption.
Their proposed methods (Single Image and Multiple Image Method) seem to be new, and their experiments suggests that for certain types of corruptions their method is effective (as measured by ECE and shown by calibration diagrams in Figure 2).

Strong points:

- Even when the type of corruption in the validation dataset is different from the corruption in the test set, the proposed method can considerably improve over existing methods.
- The proposed method, similar to other post-hoc calibration methods, can be used in combination with any model.
- The illustrations, in particular Figure 2, are well done.

Unclear/Weak points:

- The proposed method relies on set of pre-specified set of corruptions that are used to generate the validation data. Therefore, I think it is important to throughly evaluate the  impact when the corruption used for validation is different from the one in the test data.
However, results are only reported when the validation corruption is "Contrast".
Furthermore, it seems the results reported are the ones on the test data averaged over several types of corruption. However, it would be interesting to see for what types of test corruption the method work/not works.

- It would be interesting to see results when the test data is completely OOD data, like using SVHN dataset for testing predictions of a model that was trained on CIFAR-10 (see Ovadia et al, 2019).

- Some intuition/analysis should be given about the proposed method.
For example, in "Single Image Method" (i) and (ii):
My understanding is that the authors want to achieve that
if a test sample is corrupted by type A, then q_i(max) should be close to one for the calibration set which type of corruption is similar to type A.

- At least in the Appendix:
Some more formal description of how Equation (3) is calculated should be given

- The formula for computing $L_j(p_{max}(x))$ in "Single Image Method" (i) should be given.

- The notation is slightly unclear, what does the "m" in $P^{CAL, j}_m$ mean?

- I am not sure what this means:
"on each of the calibration sets determined by combinations of intensity given by: {0}, {5,0}, {5,4,0}, {5,4,3,0}, {5,4,3,2,0}, and {5,4,3,2,1,0}."

- What are sizes of each calibration set in the experiments? How many calibration sets are there?

- I am also not sure about this sentence:
"Heuristically we always want clean images in our calibration set while having different shifted means." Does it mean that you would like to have both clean images and corrupted images in the validation dataset?


Minor points:
- it should probably be $L_i(p_{max}) / ...$ and not $L_j(p_{max}) / ...$  in the equation in paragraph "Single Image Method" (i).

- It might be better to rename $P^{deploy}_m$ as $P^{test}_m$, to confirm to the standard terminology of train/validation/test data.

- in "of p_max under each of the calibration models, using the probability density",
I think it is easier to understand if "calibration models" -> "calibration sets".

---

> ### Author Response · Authors · 2020-11-17
> **Unclear/weak points addressed**
>
> > - The proposed method relies on set of pre-specified set of corruptions that are used to generate the validation data. Therefore, I think it is important to throughly evaluate the impact when the corruption used for validation is different from the one in the test data. However, results are only reported when the validation corruption is "Contrast". Furthermore, it seems the results reported are the ones on the test data averaged over several types of corruption. However, it would be interesting to see for what types of test corruption the method work/not works.
>
> We believe there may be a misunderstanding here. The type of corruption in the calibration/validation set is always different from the corruptions in the test set. For the results presented, only the 'Contrast' corruption is used in the validation set. For the test set, we use all the remaining corruptions (noise, glass blur, brightness, etc).
>
> Therefore, in Figure 1, we report the average over several types of corruptions (noise, blur, brightness, etc ) with 'constrast' corruption being used in the validation set . In Figure 3, we expand on this using box-whisker plots which allows us visualize the variation across the different types of corruptions, again with 'constrast' being used only in the validation set. We do agree that it is important to see for which types of corruption the method works or not so we have added Tables 5 and 6 in the Appendix for that matter.
>
> Moreover, we added a discussion in the Appendix of how the choice of the corruption used in the calibration set affects the results. The idea is to choose a corruption that leads to calibration sets whose accuracy slowly decreases in the presence of corrupted images at higher levels, while remaining calibrated for clean images.
>
> > - It would be interesting to see results when the test data is completely OOD data, like using SVHN dataset for testing predictions of a model that was trained on CIFAR-10 (see Ovadia et al, 2019).
>
> Thank you for the suggestion. We added this in Figure 7. We also added results for MNIST trained models evaluted on Fashion-MNIST and Not-MNIST in Figure 8.
>
> > - Some intuition/analysis should be given about the proposed method. For example, in "Single Image Method" (i) and (ii): My understanding is that the authors want to achieve that if a test sample is corrupted by type A, then q_i(max) should be close to one for the calibration set which type of corruption is similar to type A.
>
> That is precisely it. When introducing the methods in section 3.1, we added additional explanations in the revised version of the paper.
>
> > - At least in the Appendix: Some more formal description of how Equation (3) is calculated should be given
>
> We added an explanation as to how Equation (3) can be computed in practice.
>
> > - The formula for computing $L_j(p_{max}(x))$ in "Single Image Method" (i) should be given.
>
> The previous presentation allowed for a more general definition of the method where $L_j(p_{max}(x))$ can be  we defined as a function of $h^{CAL,j}(x)$. Since we simply take $L_j(p_{max}(x))=h^{CAL,j}(x)$, we simplified the presentation accordingly.
>
> > - The notation is slightly unclear, what does the "$m$" in $P_m^{CAL,j}$ mean?
>
> There should be no subscript $m$, which we believe was source of the confusion. We have also simplified the notation (see answer above), hopefully it is clearer now.
>
> > - I am not sure what this means: "on each of the calibration sets determined by combinations of intensity given by: {0}, {5,0}, {5,4,0}, {5,4,3,0}, {5,4,3,2,0}, and {5,4,3,2,1,0}."
>
> It means we have a total of 6 calibration sets. The set {0} refers to the calibration set with clean images only, {5,0} with clean images and corrupted images with intensity 5, and so on. Given the feedback provided by the referees, in the new version we adopt a much simpler and intuitive choice: {0}, {0,1}, {0,2}, {0,3}, {0,4}, {0,5}.
>
> > - What are sizes of each calibration set in the experiments? How many calibration sets are there?
>
> There are a total of 6 calibration sets. The one containing only clean images, {0}, has 5000 images, while the others {0,1}, {0,2}, {0,3}, {0,4}, {0,5} have 10000 images each: 5000 clean images (the same as in {0}) and their corrupted counterparts. This is true for both CIFAR-10 and ImageNet. This information can be found in section 3.2 of the paper.
>
> > - I am also not sure about this sentence: "Heuristically we always want clean images in our calibration set while having different shifted means." Does it mean that you would like to have both clean images and corrupted images in the validation dataset?
>
> Yes, precisely! Without any clean images, we would obtain poorly calibrated methods for in-distribution images. By having both, the method remains well-calibrated for in-distribution images, and its calibration for out-of-distribution images is improved.

---

> ### Author Response · Authors · 2020-11-17
> **Minor points addressed**
>
> > - it should probably be $L_i(p_{max})/...$ and not $L_j(p_{max})/...$ in the equation in paragraph "Single Image Method" (i).
>
> That is correct.
>
> > - It might be better to rename $P_m^{deploy}$ as $P_m^{test}$, to confirm to the standard terminology of train/validation/test data.
>
> We agree and made the change.
>
> > - in "of p_max under each of the calibration models, using the probability density", I think it is easier to understand if "calibration models" -> "calibration sets".
>
> Changed.

---

### Official Review · AnonReviewer1 · 2020-10-27
**Post-hoc calibration looks promising, but many open questions remain about applications and generalization**

**Rating:** 4
**Confidence:** 4

**Review:**

In this work, the authors propose a post-hoc calibration method for potentially OOD data that relies on estimation of the "degree" of corruption for new test data. The rely on the benchmark provided in Ovadia et al. as a basis for their analysis. Ovadia et al. assessed common measurements of uncertainty such as Brier score, ECE, and entropy over a variety of datasets including MNIST + translations/rotations, CIFAR-10 and CIFAR-10C, and ImageNet and ImageNet-C for a variety of models such as vanilla neural networks, SVI, ensembles, and Dropout. By using corrupted versions of these common datasets, Ovadia et al. could evaluate how uncertainty estimates vary under dataset shift. In this work, the authors aim to improve the calibration of probabilities obtained from these models. They start by establishing a calibration set where they derive $p_{correct}$ from a sample of $p_{max}$. Then, depending on how many test images they are evaluating, they use a single image or multiple image method to attempt to determine which calibration set (of which there can be many depending on the number of corruption levels considered), the test images are closest to. Then they "correct" the model's probability estimate by weighting over the calibration sets. They show on CIFAR-10 and ImageNet that their method results in lower ECE over varying levels of corruptions.

Strengths:
* This method could be applied post-hoc to a variety of models
* Seems fast to compute
* Results in better calibration estimates

Weaknesses:
* The terminology of the paper is ill-defined. For instance, I don't think the authors ever explicitly define $p_{max}$. I understood it by context, but the presentation could be much clearer.
* When is it clear to use the "Multiple Image Method"? How can one be sure that $\{x_1,...x_m\}$ come from the same distribution?
* It was not explained why "contrast" was chosen as the calibration set.
* How can this method be applied when the OOD corruption is very far away from "contrast"? Could you evaluate how your method performs on corruptions such as translations and rotations?

Ultimately, I don't think that this method was presented in a clear enough fashion or has sufficiently demonstrated an ability to generalize to new types of corruptions and am rating this a 4 because of these reasons.

---

> ### Author Response · Authors · 2020-11-17
> **Open questions addressed**
>
> > - The terminology of the paper is ill-defined. For instance, I don't think the authors ever explicitly define pmax. I understood it by context, but the presentation could be much clearer.
>
> We now define pmax when is first mentioned in the introduction and recall its definition when introducing ours methods. We also renamed $P_m^{deploy}$ as $P_m^{test}$, as suggested by Reviewer 4 , to conform to the standard terminology of train/validation/test data. Moreover, we also rewrote the introduction.
>
> > - When is it clear to use the "Multiple Image Method"? How can one be sure that x1,...xm come from the same distribution?
>
> We envision that the Multiple Image Method to be used when it is a priori known that $x_1,\ldots,x_m$ come from the same distribution. This in fact is the assumption in the unsupervised domain adaptation where the goal is to transfer a classifier trained on a source distribution $p$ to a target distribution $q$, which is assumed to be known (see also our response to Reviewer 2). Without that assumption a statiscal test could be performed, although that falls outside of our expertise. A simple naive approach could be to randomly separate the images into two sets, construct the empirical distribution function for each and compute the Kolmogorov-Smirnov statistic.
>
> > - It was not explained why "contrast" was chosen as the calibration set.
>
> We added a discussion in the Appendix of how the choice of the corruption used in the calibration set affects the results. The idea is to choose a corruption that leads to calibration sets whose accuracy slowly decreases in the presence of corrupted images at higher levels, while remaining calibrated for clean images.  We found the results were of similar quality for most choices of calibration set, with some exceptions e.g Glass Blur corruption which has a comparatively strong corruption, decreasing accuracy significantly even at low intensities.
>
> > - How can this method be applied when the OOD corruption is very far away from "contrast"? Could you evaluate how your method performs on corruptions such as translations and rotations?
>
> We are limited here to the data available in the benchmark (Ovadia et al (2020)), but we were able to add translation a possible corruption for CIFAR10 models, for which both the single and multi image method performs well (see Table 6 in the Appendix which shows how the method performed across the different corruptions, in particular translations). We also evaluate the confidence of CIFAR10 trained models on the entirely OOD dataset SVHN (see Figure 7). Finally, we also present results on MNIST trained evaluated on entirely OOD datasets: Fashion-MNIST and Not-MNIST (see Figure 8).

---

### Official Review · AnonReviewer3 · 2020-10-28
**Simple method, provides improvements to calibration**

**Rating:** 6
**Confidence:** 3

**Review:**

The submission proposes a very simple and seemingly effective method for improving uncertainty estimates of predictions for corrupted data. The main idea is to calibrate predictive confidences assuming access to an “exposure” set of corruptions. In particular, the paper uses contrast-corrupted data to calibrate predictive confidences and shows improvements for the types of corruptions discussed in [1] (leaving out contrast).

(+) The improvements seem to be fairly consistent, and the method is simple and intuitive. It is interesting to know that such improvements can be had, i.e. calibration on one type of corruption is transferrable to other types to some extent.

(-) While the 'single image method’ is more or less intuitive, the motivation for the ‘multiple image method' is less clear. For example, why not compare the distribution of p_maxes in S^deploy to the distribution of p_maxes in each p^CAL? Or, why not do a similar weighted averaging as in the single image method, with divergences between these two distributions providing the weights?

(-) While this paper mostly builds on prior work, it would be interesting to see calibration advantages on more than just the artificially corrupted sets. The prior work [2] that this submission builds upon reports results across a fairly wide range of tasks and out-of-distribution types. It would be more interesting to see if such calibrations with a specific set can be relevant for more realistic OOD cases as well, as in [2].

The sentence "Heuristically we always want clean images in our calibration set while having different shifted means” is not clear, could the authors elaborate? Why might we prefer such a calibration set? This might be an important point for the reader, since it motivates the particular choice of contrast-corruption.

An obvious baseline would be to perform the same recalibration procedure, but with the validation set (i.e. non-corrupt data), to figure out if corrupt sets in particular are required for calibration. I suspect they are, since performance at such sets are likely to be poorer, which would allow for more calibration room over a larger “error”-space.

Overall I think this paper could be interesting in that it informs us of the possibility of transferring recalibration from a corruption-exposure procedure, which to my knowledge is a novel reporting. More experiments as described above would improve the paper, by making a more compelling case for such exposure-based recalibration techniques.

[1] Benchmarking neural network robustness to common corruptions and perturbations, Hendrycks and Dietterich

[2] Can you trust your model’s uncertainty? Ovadia et al.


Post Rebuttal:
Thanks for the response, and the new experiments. I continue to think that this is a nice simple method that works well enough to be interesting. I retain my initial rating.

---

> ### Author Response · Authors · 2020-11-17
> **Additional experiments added**
>
> > (-) While the 'single image method’ is more or less intuitive, the motivation for the ‘multiple image method' is less clear. For example, why not compare the distribution of p_maxes in S^deploy to the distribution of p_maxes in each $p^{CAL}$? Or, why not do a similar weighted averaging as in the single image method, with divergences between these two distributions providing the weights?
>
> In fact, we do compare the distribution of the pmax values of $S^{deploy}$ (in the revised version we use the notation $S^{test}$ instead per suggestion of Referee 4) but we do so based solely on the mean. We tried using the KL-divergence but obtained comparable results and therefore kept the mean for simplicity. Moreover, one can still interpret the method as a weighted average but with $q_i \equiv 1$ and $q_j \equiv 0$ with $j\neq i$,  where $i$ denotes the calibration set with pmax mean closest to the pmax mean of the test set.
>
> > (-) While this paper mostly builds on prior work, it would be interesting to see calibration advantages on more than just the artificially corrupted sets. The prior work [2] that this submission builds upon reports results across a fairly wide range of tasks and out-of-distribution types. It would be more interesting to see if such calibrations with a specific set can be relevant for more realistic OOD cases as well, as in [2].
>
> While the methods proposed here can be for instance extended to text categorization task described in [2], our focus in this paper is image classification. In order to test our method on more realistic OOD cases, we added results of CIFAR10 trained models evaluated on SVHN and MNIST trained models evalutated on FASHION-MNIST and NOT-MNIST. See Figures 7 and 8. Compared to [2], the improvements are significant.
>
> > The sentence "Heuristically we always want clean images in our calibration set while having different shifted means” is not clear, could the authors elaborate? Why might we prefer such a calibration set? This might be an important point for the reader, since it motivates the particular choice of contrast-corruption.
>
> Previous methods were well-calibrated on uncorrupted images, but poorly calibrated (overconfident) on corrupted images.  Our contribution is to improve calibration on both uncorrupted and corrupted images.  In order to do so, we need have various levels of corruption in our calibration sets.  By always having clean images in our calibration set we keep the ratio of clean and corrupted images the same.  Otherwise, our method would be under-confident on uncorrupted images.
>
> > An obvious baseline would be to perform the same recalibration procedure, but with the validation set (i.e. non-corrupt data), to figure out if corrupt sets in particular are required for calibration. I suspect they are, since performance at such sets are likely to be poorer, which would allow for more calibration room over a larger “error”-space.
>
> We added this in the Appendix. As expected, the inclusion of the corrupted images in the validation set is needed, in particular for higher levels of corruption intensity.

---

### Official Review · AnonReviewer2 · 2020-10-29
**Clearly flawed approach relies on extra information**

**Rating:** 5
**Confidence:** 4

**Review:**

This paper studies the problem of providing calibrated predictions for out-of-distribution data. They propose algorithms for both calibrating predictions given a single image from the unknown distribution as well as given multiple images from the unknown distribution. They propose an algorithm that estimates which “calibration distribution” the novel image came from, and then use calibrated predictions for this distribution. They evaluate their approach on a standard image datasets including CIFAR-10 and ImageNet, and show that their approach outperforms existing work.

Pros
- Important problem

Cons
- The approach claims to work on out-of-distribution data, but assumes the possible novel distributions are known
- Missing related work

The approach proposed by the authors is fundamentally flawed: while they do not directly assume to know which “unknown distribution” the novel image is from, they assume it is from one of a small set of possibilities. This information is not assumed in existing work, and fundamentally alters the problem, making it simple to address and uninteresting.

The proposed approach is also very simplistic, which is not a flaw in and of itself but is a consequence of the extra knowledge they assume. In particular, given this extra information, the authors simply predict which shifted distribution the novel example is from, and then use the calibrated prediction for that distribution.

In practice, this problem is important for handling unanticipated distribution shifts in production. If the distribution shift is known and anticipated, then a much more natural approach would be to simply use data augmentation to generate data from the shifted distribution and train the model on this extra data.

In addition, there recent work in this area that the authors do not cite, for instance:

Park et al., Calibrated Prediction with Covariate Shift via Unsupervised Domain Adaptation. In AISTATS 2020.

Wang et al., Transferable Calibration with Lower Bias and Variance in Domain Adaptation. In NeurIPS 2020.

-------------------------------------------------------------------------------------------------------------------------------

Post rebuttal: I have updated my score based on the clarification provided by the authors. My remaining concern is that I still think the baselines considered by the authors is incomplete. In particular, the calibration under distribution shift techniques can still be applied, just using either just a single test image or their set of multiple test images. Admittedly, this approach would probably not perform well for a single image, but in Table 5, it seems like oftentimes multiple images are needed to even beat Ovadia et al. (2019).

---

> ### Author Response · Authors · 2020-11-17
> **Clarification on problem solved**
>
> First, we would like to thank you for taking the time to read our paper and for the suggested references, which we have added to the related work section, together with a few more.
>
> > The approach proposed by the authors is fundamentally flawed: while they do not directly assume to know which “unknown distribution” the novel image is from, they assume it is from one of a small set of possibilities. This information is not assumed in existing work, and fundamentally alters the problem, making it simple to address and uninteresting.
>
> Upon reading the suggested references, we believe you have in mind the problem of calibration in the *unsupervised domain adaptation* setting, whereas what we are doing is *calibration under distribution shift*. Here's our understanding of each.
> In *unsupervised domain adaptation*, the goal is to calibrate a model, trained on the source distribution $\rho_{train}$, to the target distribution $\rho_{test}$, given labeled examples from the source distribution and unlabeled examples from the target distribution. An example would be to train a model on MNIST and evaluate it on SVHN. In *calibration under distribution shift*, we are looking at corrupted images from the same data set, with unknown corruptions (see Tables 5 and 6 in the revised version for all the corruptions considered). The confusion here may arise because our multi image method can also be applied to unsupervised domain adaptation (although this was not our focus). We think this is a natural mistake to make and we have therefore rewritten the introduction to make it clear which problem we are addressing. We kindly ask that you please take a second look with this distinction in mind.
> Moreover, we should also point out that our single image method tackles the much harder problem of calibrating on a single image (without even making use of unlabeled examples of the target distribution).
> In the regards to the use of extra information, we don't believe that to be the case. As AnonReviewer3 explains in the first paragraph of his review our paper "uses contrast-corrupted data to calibrate predictive confidences and shows improvements for the types of corruptions discussed in [1] (leaving out contrast)". This is a crutial point. While the contrast corruption is used to build the surrogate calibration sets, the methods are evaluated at test time on never seen corruptions, hence why we refer to them as out-of-distribution.
> To further address your concerns, in the revised version of the paper we evaluate our method on completely OOD data: CIFAR-10 trained models evaluated on SVHN and MNIST trained models on Fashion-MNIST and Not-MNIST (see Figures 7 and 8).  Our methods are very effective in detecting OOD data.
> Finally, we point out that, based on the references you provided, a more compelling experiment would be to measure the performance of our methods using MNIST trained models evaluated on SVHN. However, the benchmark provided by Ovadia et al (2020) requires additional work to run such an experiment, (we just used the model outputs provided, and these are not available on SVHN for MNIST trained models).  Unfortunately we do not think we will have the time to run such experiments that fall under the unsupervised domain adaptation setting during this short rebuttal period. We may be able to present some restricted experiments if you still deem it necessary and worthwhile given the different problem we propose to solve.
>
> > The proposed approach is also very simplistic, which is not a flaw in and of itself but is a consequence of the extra knowledge they assume. In particular, given this extra information, the authors simply predict which shifted distribution the novel example is from, and then use the calibrated prediction for that distribution.
>
> Again, we cannot help to think that there may be a misinterpretation of what we are doing here. We emphasize once again that we calibrate on contrasted images, and at test time, the OOD images presented are shifted by other different transformations (e.g. gaussian blur, various forms of noise, etc.).  These have not been seen by the model prior.

---

### Author Response · Authors · 2020-11-17
**Revision Submitted and general remarks**

Dear reviewers,

Thank you for your time and remarks. For your convenience we highlighted all the changes in the pdf in blue. We have addressed your comments and concerns by replying individually to each one of you and we hope the additional explanations and experiments have addressed your previous questions.

We would like to emphasize the following:

- Our proposed methods are indeed remarkly simple and rely on a purely statistical approach, requiring no additional training. The cost of the methods relies solely on binning the data into histogram thus making it very fast to compute. We shared the code in the suplemental material.
- We do not rely on extra information to solve the problem. Only one corruption is used to form the calibration sets (contrast is used in the paper and in the revised version of the paper we include in the Appendix a discussion regarding use of other corruptions). At test time, we evaluate the calibration of the models on corruptions (brightness, glass blur, speckle noise, etc) never seen by the  model. This is an important point that we believe may be the source of some misunderstanding regarding our method. In the revised version, we made sure to emphasize this point throughout the paper.

Please let us know if any further clarifications would help you reconsider your rating.

---

### Author Response · Authors · 2020-11-24
**Final Revision**

Dear Reviewers and Area Chair,

We have made a final revision on our paper. For your convenience we highlighted all the new changes in red, with the changes in the first revision in blue.

In particular, we would like to draw your attention to the new Figures 3 and 11 and the extra discussion in the Appendix. We hope that these not only help explain how the methods work, but also why they work. We also updated Figure 9, as we notice we did not compile the correct figure in our latex code. While the changes are almost negligible and the conclusions remain unchanged, we felt we should still mention it.

We hope these additional explanations, as well as the previous ones, help you reconsider your ratings, like it did for Reviewer 3 after our first revision. Using a purely statiscal approach, we have proposed a simple and intuitive method (**R3**) that is fast to compute (**R1**), requiring no training of the Neural Net classifier, and that consistently improves calibration across different corruptions (**R1**,**R3**,**R4**).

---

### Decision · Program_Chairs · 2021-01-07
**Final Decision**

**Decision:**

Reject

**Comment:**

This paper presents a method to improve the calibration of neural networks on out-of-distribution (OOD) data.

The authors show that their method can be applied post-hoc to existing methods and that it improves calibration under distribution shift using the benchmark in Ovadia et al. 2019.

However, reviewers felt that the theoretical justification for why this works is unclear (see detailed comments by R1 and R4), and some of the choices are not well-justified. Revising the paper to address these concerns with additional theoretical and/or empirical justifications should improve the clarity and strengthen the paper.

I encourage the authors to revise and resubmit to a different venue.